# Automated Unsupervised 3D Tool-Path Generation Using Stacked 2D Image Processing Technique

**Tadele Belay Tuli** [1],* and **Andrea Cesarini** [2]

[1]   Department of Electromechanical Engineering, Addis Ababa Science and Technology University, P.O. Box, Addis Ababa 16417, Ethiopia
[2]   Andrea Cesarini, Istituto Nazionale di Fisica Nucleare (INFN), Sezione di Firenze, Via B. Rossi, 1, I-50019 Sesto Fiorentino, FI, Italy; andrea.cesarini@fi.infn.it
*   Correspondence: tadele.belay@aastu.edu.et

**Abstract:** Tool-path, feed-rate, and depth-of-cut of a tool determine the machining time, tool wear, power consumption, and realization costs. Before the commissioning and production, a preliminary phase of failure-mode identification and effect analysis allows for selecting the optimal machining parameters for cutting, which, in turn, reduces machinery faults, production errors and, ultimately, decreases costs. For this, scalable high-precision path generation algorithms requiring a low amount of computation might be advisable. The present work provides such a simplified scalable computationally low-intensive technique for tool-path generation. From a three dimensional (3D) digital model, the presented algorithm extracts multiple two dimensional (2D) layers. Depending on the required resolution, each layer is converted to a spatial image, and an algebraic analytic closed-form solution provides a geometrical tool path in Cartesian coordinates. The produced tool paths are stacked after processing all object layers. Finally, the generated tool path is translated into a machine code using a G-code generator algorithm. The introduced technique was implemented and simulated using MATLAB® pseudocode with a G-code interpreter and a simulator. The results showed that the proposed technique produced an automated unsupervised reliable tool-path-generator algorithm and reduced tool wear and costs, by allowing the selection of the tool depth-of-cut as an input.

**Keywords:** tool path; 3D modeling; CAD/CAM; image processing; G-code

## 1. Introduction

Computer numerical controlled (CNC) machinery opened a window, decades ago, for automating the machining system. The primary function of the CNC machine tool is to execute a sequence of multi-axis motions following part geometry [1]. At present, digital manufacturing is transforming the method of generating and simulating the tool-path movement of automated and computerized machinery tools. Particularly, computer-aided design (CAD) and computer-aided manufacturing (CAM) tools are commonly used to simulate and visualize what the real-time operation would look like. Safe, optimal, and accurate machining processes demand for automated, robust but low-intensity computation tool-path planning, to lower machinery vibration, tool wear or breakage, and thermal deformation of the machine tools [2]. In this context, much work is being done in order to allow the machinery tools to perform numerous complex and constrained tasks [3–6] in the optimization of cutting-tool parameters, such as feed rate, cutting speed, depth-of-cut, and feeding distance [7,8], which are provided as input. On the other hand, image processing techniques are mainly used for recognition, detection, and tracking objects [9]. In CAM systems, tool-path generation is mainly implemented either in design and modeling tools (e.g., Solidworks®) or tool-path generation tools

(e.g., Mastercam®); where, unfortunately, time is wasted while transcoding file formats. Moreover, a STEP based tool-generation approach was implemented in [10], to minimize the file exchange time.

The present work was aimed to present a reliable, automated, and unsupervised technique to generate tool paths for numerically controlled machining tools, by slicing 3D models into 2D layers. Each layer was discretized and converted to a binarized image. In each 2D layer, a start point was selected, a tool path was generated in the Cartesian coordinates between the image contiguous pixels and finally, it was translated into a machine language like G-code.

## 2. State of the Art

Recently, realistic modeling, together with the optimization of the production process, were paired together in order to analyze the process parameters for a minimum cycle time [5,10]. This enables numerically controlled devices—such as CNC machine tools, 3D printers, robots, 3D scanners, and coordinate sampling machines—to successfully perform their desired task [11–14]. In [15], contour parallel tool-path optimization was presented for the milling operation of 2D pocket regions. In this work, the pocket boundary of an object was converted into a binary image to extract the tool-path information. Multiple cutters and hybrid tool-path patterns were considered in [2], for complex islands and curvilinear pockets, by introducing the machining time as a constraint. Moreover, a high accuracy could be obtained by using continuous-motion data and a small depth-of-cut. In this context, time and material waste is a crucial factor to be considered. Reference [16] presented an automatic G-code generating approach for the milling process, based on the MATLAB environment. In [17], a new geometry was represented for the CAD/CAM for a highly parallel scalable environment. In this method, parallel computing was considered to solve big computational problems that are usually related to the CAM process. In general, computational time, tool path-generation cost, and virtual simulation environments are barriers between researchers and small companies. Automated machine tools require a concept for process realization of manufacturing processes. According to [18,19], models such as STL could simplify path-generation difficulty at the early stages. Similarly, feature-detection using a classical approach from a binary image is also well-investigated [20]. However, there is a lack of a common and generalized approach for automatic generation of a tool-path based on any 3D model, without any human supervision. Existing solutions either are coupled to CAD tools and are expensive or are not so easy to be applied. Therefore, adoption of a simple, computationally low-intensity but accurate path-generation approach is advisable. Notably, for small companies that need to verify what their product would look like, before the actual realization.

## 3. Materials and Methods

CNC machinery tools operation processes are usually defined using CAD/CAM tools. In this phase, product design, and manufacturing processes were defined by following the functional requirements. In this section, we present the materials and methods utilized to realize the concept of design for the manufacturing processes. Figure 1 shows the concept of a tool-path generation approach based on a 3D model design. The approach considered a 3D CAD model of the final product to generate a tool-path using a predefined billet size. Depth-of-cut, cutting speed, cutting length, and number of passes were the parameters considered.

According to Figure 1, we considered two stages for the design and workflow of this research. Geometric modeling deals with product design and functional features identification. Whereas analytical modeling is the analytical and logical processing of the CAD model using analytical and logical approaches.

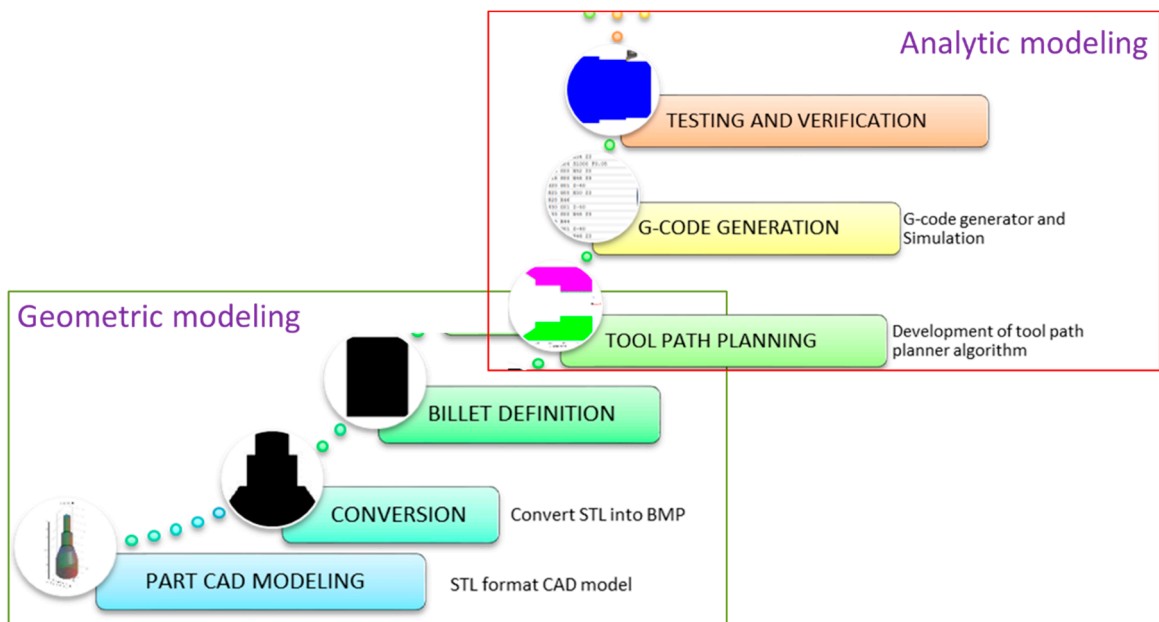

**Figure 1.** Diagram of research design and workflow. In this workflow, a geometric model was imported into a simulation environment and processed into numerical values.

### 3.1. Geometric Modeling and Parameter Identification

CAD is the use of computer systems to assist the creation, modification, analysis, and optimization of a geometric design [16]. Various CAD software packages used data formats such as interchangeable graphics exchange (IGES), standard for the exchange of product model data (STEP), stereolithography (STL), drawing exchange format (DXF), and object file format (OFF). These data exchanging formats convey geometric data in different structures as well as varying in memory size, vertices, and facet numbers. We compared how these parameters vary using a constant cube of 10 mm size (see Figure 2 and Table 1). In Table 1, the STL file format is simple for computation.

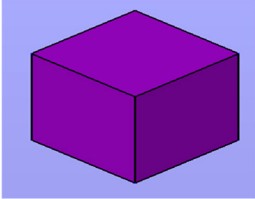

**Figure 2.** A cube model ($10 \times 10 \times 10$ mm$^3$).

**Table 1.** Comparison of different CAD file formats

| Parameter | STEP | STL | IGES |
|---|---|---|---|
| Memory (Bytes) | 16.7 K | 684 | 21.3 K |
| 3D | Yes | Yes | Yes |
| Vertices | 18 | 36 | 18 |
| Faces | 6 | 12 | 6 |

Most CAD modeling software supports standard tessellation language (STL). It is a representation of triangulation in the 3D surface, where three vertices describe each facet (see Figure 3). STL file format supports only surface features like colors, textures, scales, and units. The file formats are arranged as a combination of facets, vertices, and edges. Generating the tool path based on STL file format is

initially challenging as vertices are located in a disordered way and demand further processing in order to sort and rearrange—as it is illustrated in Figure 3 (for a cube has 36 vertices and 12 facets).

$$
\text{Vertices} =
\begin{bmatrix}
0 & 0 & 10 \\
0 & 0 & 0 \\
\vdots & \vdots & \vdots \\
10 & 10 & 10 \\
0 & 10 & 10
\end{bmatrix},
\text{Facets} =
\begin{bmatrix}
1 & 2 & 3 \\
4 & 5 & 6 \\
\vdots & \vdots & \vdots \\
31 & 32 & 33 \\
34 & 35 & 36
\end{bmatrix}
$$

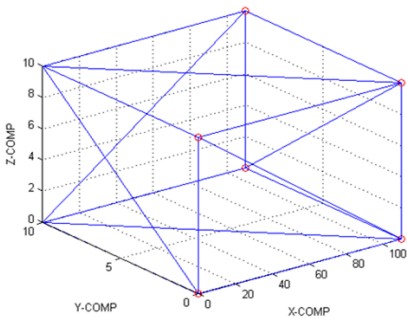

**Figure 3.** A cube with vertices and facets indices. A triangulation of vertices is represented using edges, and these combinations can be given in matrices form.

Geometric modeling of 3D modeling is performed using CAD tools and the file is imported to MATLAB® software by using STL file reader. The imported model was re-sampled and discretized into (n, m, k) dimension (see Figure 4).

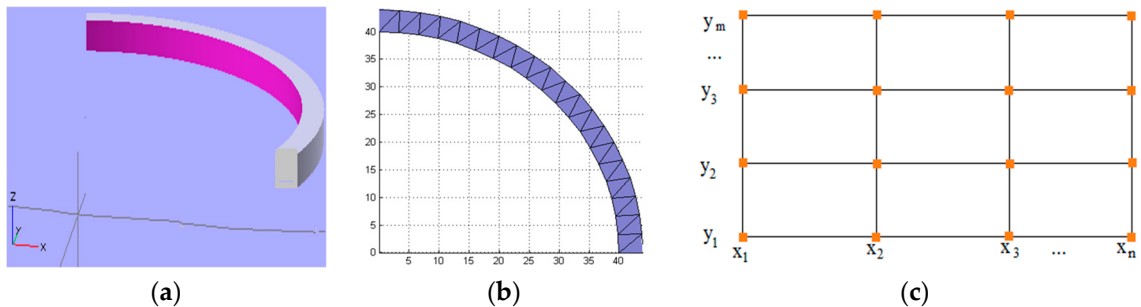

**Figure 4.** (**a**) CAD model in STL file. (**b**) 3D model in MATLAB environment. (**c**) The discretization approach.

### 3.2. Model Segmentation and Reconstruction

Essential features such as three-dimensional sizes, segmentation size, and model logics are identified to create a slicing algorithm [21,22] which recursively slice a model into several pieces. This algorithm is used to map the sliced layer with cutting depth. The slicer is used to create a two-dimensional layer model. If multiple tool pass is desired, the slicer will be configured accordingly. Therefore, a generic approach is implemented to segment the model into finite segments. In this process, each segment is localized by their respective position and orientation as point clouds. A line segmentation that is given by Equation (1) takes the maximum and minimum values from the model. We applied three times to generate point clouds for all orientations (e.g., 3D space).

$$
S_{N+1} = S_N + \frac{S_n - S_1}{n - 1} \tag{1}
$$

where N and n are the current point and number of segmentations, respectively. Here, 'S' represents $x$, $y$, and $z$. After the segmentation of an object, we determined if the vertices of the segmentation lie inside or outside a 3D model using binary logic. We created a container that has a specific position and orientation and assigned a Boolean logic at each element. Figure 5 elaborates this approach. In this figure, an original model (Figure 5a) contains enclosing grids. The spatial representation is shown in Figure 5b. Moreover, Figure 5c,d displayed the output of the voxelized space. This stage is, a reconstructed model after we performed the slicing and triangulation processes.

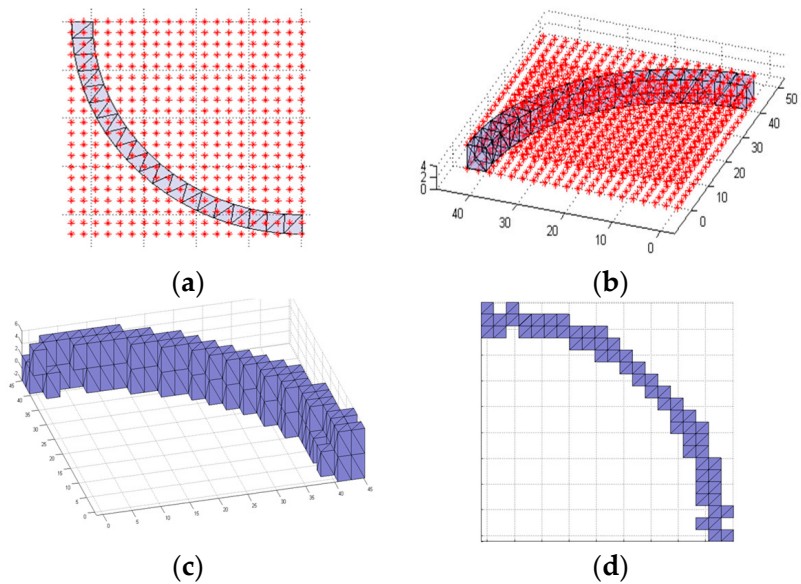

**Figure 5.** A model contained in a grid (**a**) 2D; (**b**) 3D; (**c**) 3D discretized and voxelized; (**d**) 2D projection of (**c**).

In order to enclose the point into the center of the cube, we applied linear translation of the coordinate points from both initial and endpoints to compress the outmost boundary of the model. Equations (2) and (3) explain how such translation is used to shift the enclosing grid.

$$x_{gi} = x_i - \frac{x_{i+1} - x_i}{2} \qquad (2)$$

$$x_{gf} = x_f + \frac{x_{i+1} - x_i}{2} \qquad (3)$$

where $x_{gi}$ and $x_{gf}$ are the initial and final grid points that are shifted from the center of cube cells [23].

According to Algorithm 1, the voxel model is created by combining neighbor triangulations. It works by checking if the neighbors lay inside or outside of the objects. If the neighbors are outside the object, the algorithm continues by drawing a boundary which finally converges to the cube cells (given in Figure 5d). By repeating the same procedure for the three directions, it creates three-dimensional arrays. The pseudocode is implemented using a MATLAB script.

The geometric accuracy of Algorithm 1 is adjusted by selecting an appropriate segmentation resolution. Increasing the sampling size will pull the cube into the center of the point. This makes the model converge into an accurate dimension. However, computation becomes intensive for complex objects. However, this can be significantly reduced by converting CAD models into two-dimensional binary images. Taking advantage of a bitmap image file, it is possible to develop a simplified path planner algorithm. Moreover, it neglects color information. In our case, the value of color is described by 1 bit for black and white. This is the smallest possible color information that can be described by the monochrome bitmap image file. In this case, pixels with 0 values are referred to as black, whereas pixels with a value of 1 are white colors.

---

**Algorithm 1**: Pseudocode for voxelization as a binary logic

---

```
grid_data = zeros(rx,ry,rz);
P0 = Facet position
Nf = Array for normal facets
for nz = 1 : rz
     for ny = 1 : ry
          for nx = 1 : rx
               % Get the point
               p = [ xa(nx) ; ya(ny) ; za(nz) ];
               % Find the closest Facet
               vertices_distance = ∑(([ P0(1,:)-p(1) ; P0(2,:)-p(2) ; P0(3,:)-p(3) ])²);
               [v,ind] = min(vertices_distance);
               % Add Point if it is enclosed inside an object
               data = dot(N_f(:,ind),p-P0(:,ind));
               grid_data(nx,ny,nz) = (data>=0);% logical array size of NxMxK
          end
     end
end
```

---

### 3.3. Point Cloud Generation Using Image Processing Techniques

Clouds of points carry proper information regarding position and orientation in the space. It is necessary to generate such points starting from a tessellated 3D model. Slicing a 3D model is applied to generate an array matrix of vertices and normal directions which is further converted into binary 2D images. Then, each binary scale image is provided of two-dimensional information which can be interpreted using pixel data. Pixels and geometric dimensions are used to map and transform binary image and 3D model. Figure 6 presents a conceptual functional diagram for mapping 3D model in a stacked pile of 2D images.

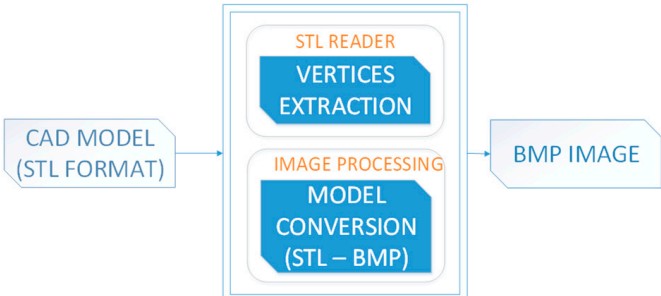

**Figure 6.** Block diagram of CAD to BMP processing. The STL reader imports CAD model into the MATLAB working space and extracts vertice, edge, and facet data. The image processing algorithm function further processes CAD data and creates a BMP output.

Converting an STL file into a BMP image file implies the transformation of vectors of facets and vertices of the part model into a scalar cloud of points. In the Figure 7, the process flow is shown. The logic template helps to identify the region of the work which has to be removed and/or has to remain during machining processes. The process of 2D discretization and image conversion is elaborated using Figure 8.

Using Algorithm 1 and a concept presented in Figure 7, the output of segmentation is shown in Figure 8.

Hence, converting the STL model into a BMP image file, on the other hand, is equivalent to converting vectors of facets and vertices of the part model into a scalar cloud of points and creates a part model which has two dimensions (see Figure 9).

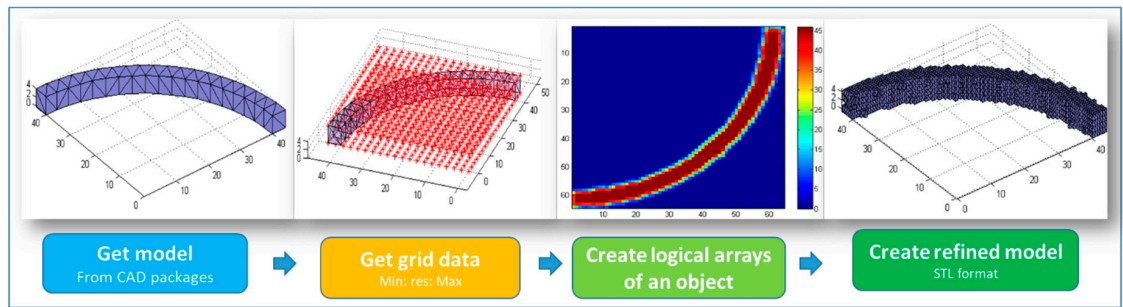

**Figure 7.** Process flow for CAD model into point cloud processes.

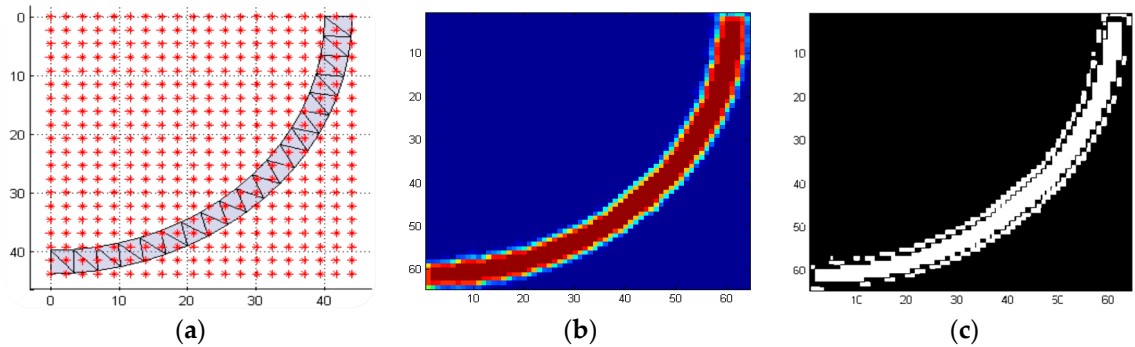

**Figure 8.** Converted CAD model from STL to BMP image file. (**a**) 2D projection of a 3D model on 2D grid container, (**b**) Intensity image for contained models, (**c**) Binary scale of image (**b**).

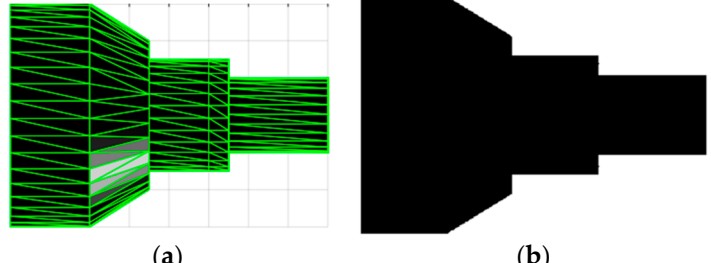

**Figure 9.** (**a**) CAD model imported as STL file format and (**b**) Converted CAD model from STL to BMP image file.

### 3.4. Tool Path Motion Parameters

Billet dimension: A billet is a length of metal that has a round or square cross-section, with an area less than 230 cm$^2$. Actually, it is essential to select the billet dimension with a minimum amount of material removal because it minimizes the machining time, power consumption, and waste of material. In this paper, the billet size is automatically computed by adding an offset value to the absolute difference of maximum and minimum value in both radial and longitudinal motions.

$$b_s = |x_{max} - x_{min}| \times |z_{max} - z_{min}| \tag{4}$$

where $b_s$ is billet size given as M × N, where M is the length of billet along longitudinal (*z*-axis) and N is the diameter of the billet which supports a motion along the radial axis (*x*-axis).

Once the dimension of the billet is determined, and the part geometry of the final product is modeled in section (CAD model), now what is left before the generation of the tool path is determining the material which is going to be removed from the work material. This portion is essential particularly to determine the depth-of-cut, number of passes, spindle speed, and feed rate.

Cutting speed: Cutting speed is the speed difference between the cutting tool and the surface of the work piece it is operating on. It is expressed in units of distance along the work piece surface per unit of time.

$$N = \frac{K \times V}{\pi \times D_0} \tag{5}$$

where $N$ is machine speed in revolutions per minute (RPM), $K$ is a constant to correct speed ($V$) and part diameter ($D_0$) units, and $V$ is desired cutting speed, a handbook value. Figure 10 shows the physical meaning of motion parameters and the corresponding symbols.

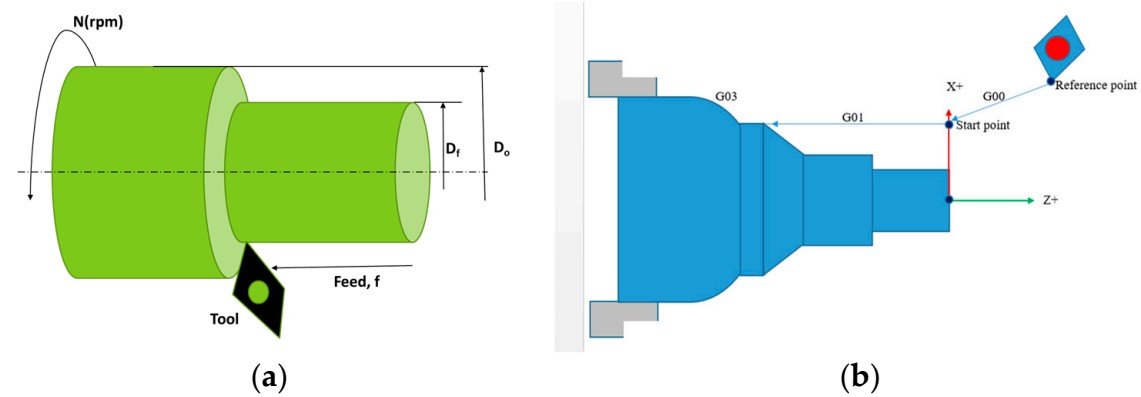

(**a**)          (**b**)

**Figure 10.** Schematic diagram of turning parameters and demonstration. (**a**) Geometric description and parameter definitions for turning operation, (**b**) Axis assignment convention and demonstration of motion types.

Cutting time: Cutting time is the time taken per operation

$$t_c = \frac{L + A}{f_r \times N} \tag{6}$$

where $t_c$ is the cutting operation, $L$ is the length of the cut, $A$ is the approach allowance distance to the starting point, and $f_r$ is the machine feed rate which is read from a handbook.

Depth-of-cut: It is the difference of material removed from the initial diameter to the final diameter

$$d = \frac{D_f - D_o}{2} \tag{7}$$

where $d$ is the depth-of-cut, $D_f$ is the final diameter, and $D_o$ is the initial diameter.

### 3.5. Tool-Path Generation and Parsing

Cutting operations can be turning, milling, drilling, or boring processes. In this work, minimal cutting and milling processes are considered. The tool-path generation typically refers to many line segments representing one linear movement of the cutter. Each linear movement, the start, and the endpoint should be accurately positioned. More line segments in tool path generation influence not only computation time but also affects the efficiency of the production process [16]. Tool-path generation by the extraction of pixel-level image information is essential to identify the part to be removed. Subtraction of two images, the billet image and the final product, yields the image of material to be removed (see Equation (8) and Figure 11).

$$I_m = I_{billet} - I_{part} \tag{8}$$

where $I_m$ is the image of the material to be removed and $I_{billet}$ is an image of billet that is user-defined and $I_{part}$ is the final product image file.

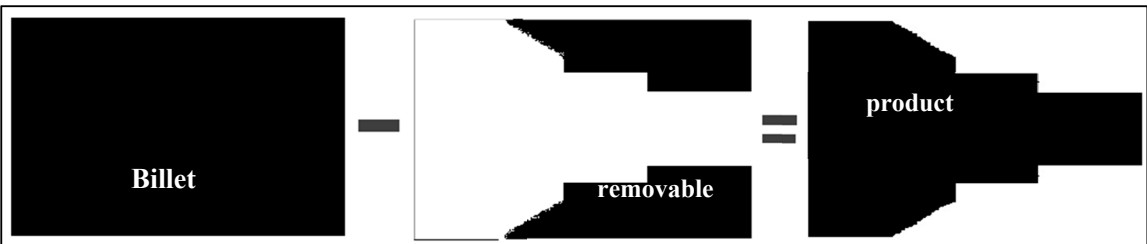

**Figure 11.** Conceptual demonstration of image subtraction technique for turning the process.

Using a mapping function between 3D model geometry and BMP images, tool path coordinate points are generated using Algorithm 2. This algorithm is implemented using the MATLAB script. For milling operation, for instance, the process is given by three-axis motion. However, one of the axes denotes depth-of-cut. Therefore, using a user-defined variable for depth-of-cut, it is possible to calibrate x- and y-components along with horizontal and vertical image pixel information. On the other hand, turning operation which is characterized only by *x*-axis and *z*-axis due to *x*- and *y*-axis symmetry can be represented with a simplified path planning method. In this situation, a case function is created to identify if the process is turning or milling operation as user input.

---

**Algorithm 2**: Pseudocode for image processed path generator

---

```
[z,x] = pixels data from image along(z,x) axis
Pcor = [];                                 % Initializing the dynamic coordinate array
k = 0;                                     % Counter
t = 0;                                     % Counter
for i = From Zo to Zf
    t = t + 1;                             % Able to count the number of tool passes
    for j = From Xf to Xo                  % Holds true for materials to be removed
        if image(i,j)>0;
            Pnew = [j,i];
            k = k + 1;                     % Counts the true pixels to be removed
        else
            Pnew = [];                     % Final product pixels
        end
        P = Pnew;
        Pcor = [Pcor;P];                   % Creates a vector of tool path
    end
end
```

---

To differentiate, in both processes—i.e., turning and milling operation—there are two-dimensional cutting processes. The remaining one dimension denotes a depth-of-cut which can be either user-defined or programmatically generated using slicing algorithms. In this paper, the main task is to develop a proof of concept that can be generalizable using a single step user-defined depth-of-cut.

The tool path planner given by Algorithm 2 generates the machine tool path for both feed and crossfeed motions. The algorithm iterates along the *x*-axis from maximum to the minimum value whereas, iterates from the minimum value to maximum value along the *z*-axis. What comes next is to develop an algorithm used to create G-code, which is readable by any CNC machine tools and simulation software packages.

*3.6. G-Code Generation*

In any CNC machine, there are three basic motions G-codes. These are: G00 for rapid point to point positioning, G01 for linear interpolation code, and G02/03 for circular interpolations.

G00—Rapid Positioning: This is the point-to-point motion used to position the cutter from one point to another without having to coordinate the velocities of any of the moving axes. It is used for the rapid positioning of the cutter without cutting.

$$x_i = \frac{x_f - x_o}{z_f - z_0}(z_i - z_o) + x_o \tag{9}$$

where $(x_i, z_i)$ are the running position of the cutter, $(x_f, z_f)$ are the desired position of the cutter, and $(x_i, z_i)$ are the reference position of the cutter.

G01—Linear Interpolation Code: This is the code that involves continuous manipulations of velocities of each axis during contour machining. At the same time, the velocities of the axes are controlled to keep the tool on a straight path in a plane of motion. By adopting constant displacement varying interpolation, which is given in [1], it is easier to develop the algorithm that interpolates the linear motions.

Let us assume that the cutting tool center is to follow the linear path shown in Figure 8 given by G00. By considering two points—i.e., the starting and end point, the time law ($T_i$)—the sampling size (N), it is possible to have the Equations (10) and (11).

$$x(k) = x(k-1) + f_x(k)T_i(k) \tag{10}$$

$$z(k) = z(k-1) + f_z(k)T_i(k) \tag{11}$$

where $f_x$ and $f_z$ are axis velocities at time interval $k$ are given by

$$f_x(k) = \frac{\Delta x}{T_i(k)}, \ f_z(k) = \frac{\Delta y}{T_i(k)} \tag{12}$$

Manipulating the time $T_i$ is, on the other hand, refers to the manipulation of the axis feeds according to the vector feed and velocity profiles. However, the incremental displacements are constant and given by

$$\Delta x = \frac{x_f - x_o}{N}, \ \Delta z = \frac{z_f - z_o}{N} \tag{13}$$

G02, G03—Circular Interpolation Codes: The velocities of two axes on a plane of motion are varied to keep the tool following the given arc at the specified feed velocity. Two types of circular interpolation commands are used in CNC systems. These are clockwise motion (G02) and counter clock motion (G03).

Assuming constant displacement varying interpolation methods (for detail derivation, refer [1]), we can compute the tool paths Cartesian coordinates using Equations (14) and (15).

$$x_{n+1} = R \times \sin(\theta_0 + n\Delta\theta + \Delta\theta) \tag{14}$$

$$z_{n+1} = R \times \cos(\theta_0 + n\Delta\theta + \Delta\theta) \tag{15}$$

where R is the radius of the arc, $\theta$ is angular position, and n is the index of a coordinate point.

For example:

- G00 X45 Y20 Z00: Rapid movement to coordinates of (45, 20, 0)
- G01 X45 Y20 Z00 F3.5: Linear movement to coordinates of (45, 20, 0)
- G02/G03 X45 Y20 R1.0: Circular motion to coordinates of (45, 20) with a radius of 1.0

## 4. Result and Discussion

Methods that we presented in this work generates tool path coordinate systems in Cartesian space based on 3D models using image processing technique. Data such as vertices and facets are extracted from

the STL file format. The vertices are rearranged in a three-dimensional array, before they are converted into a two-dimensional monochrome bitmap image. Based on the geometric configuration, either milling or turning operation shall be performed. In turning operations, we considered a symmetrical model and therefore, only half of the model is required to generate a tool-path. In milling operations, three-dimensional motion planning is required, but one axis is dedicated to depth-of-cut, which determines the layer thickness. Full functionalities of G-codes such as coordinate systems, offsets, tools, lubricants, spindle speed, units, rapid movement, controlled feed, and so on are not implemented. However, we tested motions for rapid, linear, and circular as a proof of concept. The output result from the milling process is visualized and shown in Figure 12. In this process, the CAD model provides a quarter of a circle with a square cross-sectional area. The surface is not as smooth as the 3D model. This happens due to the smaller size of the sample during the slicing and discretization process. We investigated how increasing the number of samples which is given by Equations (2) and (3) significantly improved the smoothness for the edges.

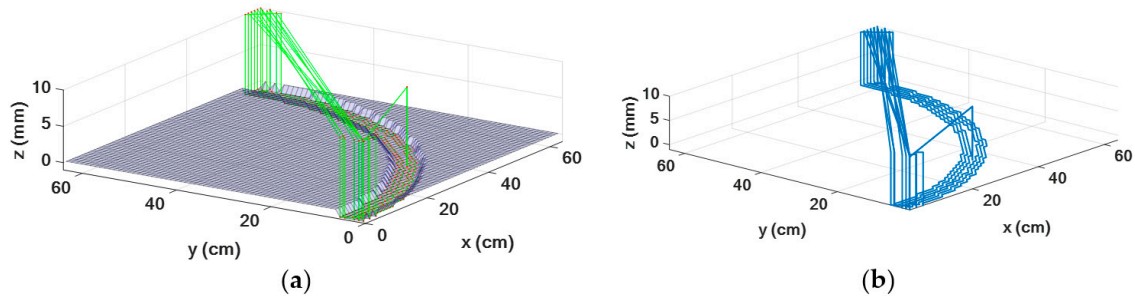

(**a**)    (**b**)

**Figure 12.** Image processing technique based milling operation simulation. (**a**) Hypersurface visualization and process definition for milling operation of circular path. (**b**) Tool-path planning and plotting.

A 3D model segmentation and reconstruction algorithm (refer Algorithm 1) returned geometrical errors for low-resolution variables. The higher sampling size, the better geometric accuracy is achieved. However, it reduced the performance of the processor for larger number of samples. In milling operation, a sequence of operation and geometric construction affected the auto-tool-path generation (see Figure 12). In this case, tool-path generation takes place along $x,y$-plane where the $z$-axis is dedicated to depth-of-cut.

The turning operation, unlike milling, a workpiece has a symmetric feature. This makes the tool-path generation and planning easier. Only path generation for depth-of-cut and feeding which are along $x$- and $z$-axes respectively are required. The output of Algorithm 2 is described using Figure 13. In this figure, tool-path motion for very small depth-of-cut is shown. Still, accuracy is affected by the sampling size. The tool-path which is shown here is the region of a part to be removed in which a machine tool is in contact. Finally, the motion is translated into machine language (G-code). We simulated the whole process using MATLAB® and we adopted the source code of [16] for turning and milling operations.

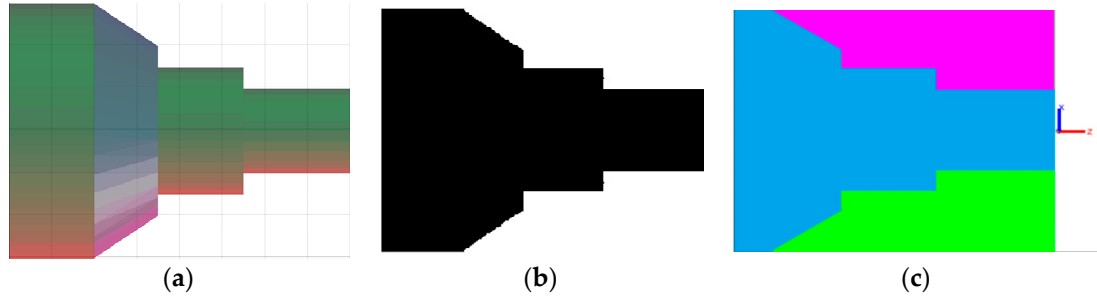

(**a**)    (**b**)    (**c**)

**Figure 13.** Image processing technique based turning operation simulation. (**a**) 3D model (front view), (**b**) 2D binary image, (**c**) path generated model).

We measure the accuracy of the generated tool-path by comparing to the 3D model. A carbide cutter tip with a length of 10 mm is geometrically modeling using surface filling in order to realize by simulation how the process takes place. In this regard, only the half model is simulated, particularly for turning operation using linear interpolation. Errors caused during the transformation of the 3D model into a 2D binary image affected the quality of the motion. However, this can be still improved by optimizing parameters such as depth-of-cut, segmentation quality, and filtering techniques. Moreover, the result can be qualitatively described using visual observation as it is shown in Figure 14.

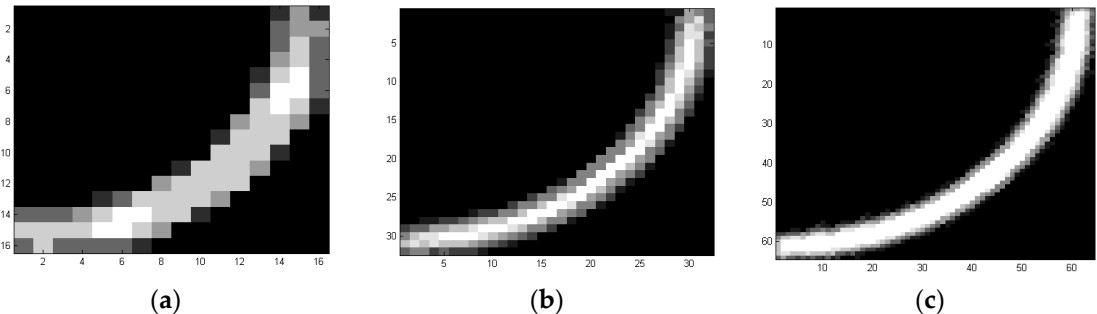

(**a**)          (**b**)          (**c**)

**Figure 14.** How accuracy of models can be affected by segmentation size. (**a**) 16 divisions, (**b**) 32 divisions, (**c**) 64 divisions.

We segmented the model into 32,000 nodes, to obtain high accurate path motion. This process yields a highly accurate turning operation which has a cutting depth of 180/32,000 mm size. In this regard, the motion is computationally costly. During the simulation, this takes up to $32,000 \times 32,000 \times 1$ milliseconds. However, this can be re-configured to different depth-of-cut if we do not consider regular segmentation in both horizontal and vertical planes.

Figure 15 shows the visualization of the tool path (a) for half model and (b) the process of tool pass. This clearly shows, a minimum depth-of-cut can be achieved if the geometry of the product is not complicated. In general, the presented approach yields an automated approach to improve cost and production error.

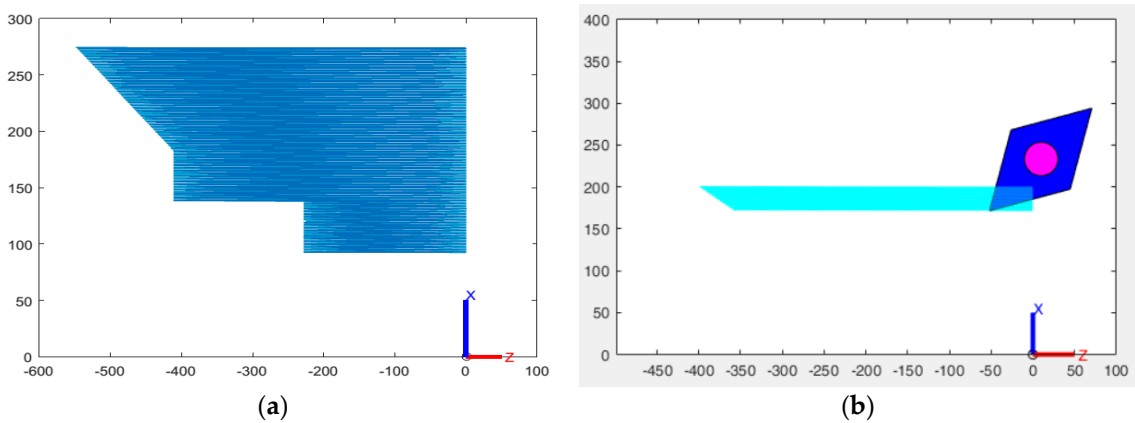

(**a**)          (**b**)

**Figure 15.** Simulation for the turning process using MATLAB script; (**a**) Path visualization for tool pass; (**b**) Simulation processes for turning operations.

## 5. Conclusions and Future Work

For noncomplex geometry, image processing based tool-path generation has the potential to simulate and visualize machining tool operations. Processing STL file format is selected due to its simplicity and reliability to generate and parse tool path coordinate points. Two different manufacturing processes, such as milling and turning operations, are considered to develop the concept of automated

tool-path generation using multilevel processing. In general, parametric model segmentation, point cloud generations from the segmented and sliced model, binary logic container creation, 2D projection, and transformation of the 2D projected model into the binary image are the main procedures we applied. In the meantime, production process variables and parameters are formulated and abstracted into the main algorithm to generate a machine-readable language like G-code. The result showed, the accuracy of the path is affected by segmentation and 3D to 2D conversion processes. The grid container which has a logic to discriminate a part of an object which is enclosed inside the object depends on sample size. The higher the sample, the better the result, but the slower the process. This research will benefit both academia and smaller companies to realize a more reliable and smaller depth-of-cut. In future work, higher-order motion generation techniques should be investigated for jerk and undesired vibrations during the cutting process.

**Author Contributions:** This research was initially conducted at the University of Trento, Italy, and extended at Addis Ababa Science and Technology University, Ethiopia. For this research article, initial writing is done by T.B.T. under the supervision of A.C.

**Funding:** This research received no external funding.

**Acknowledgments:** The authors would like to acknowledge the University of Trento, Italy, for providing research facilities.

**Conflicts of Interest:** The authors declare no conflict of interest.

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
