# Peer review of "Automated Unsupervised 3D Tool-Path Generation Using Stacked 2D Image Processing Technique"

_jmmp, doi:10.3390/jmmp3040084_

Round 1

Reviewer 1 Report

Minor edits:

Line 143  Equations (2) and (3)

Line 150 Figure 5(d)

Line 170 "If we are able to generate..."

Author Response

Dear Reviewer,

I thank you for your valuable comment. We have positively considered your comments and extensively revised the article. 

Line 143  Equations (2) and (3)

Appreciated and considered

Line 150 Figure 5(d)

Appreciated and considered

Line 170 "If we are able to generate..."

Appreciated and considered

The whole text is revised for grammer and styles. 

Kind regards

Authors

Reviewer 2 Report

This paper presents a tool path generation algorithm based on 2D images. 

Suggestions for improvement:

- A section which clearly explains the State-Of-The-Art in this field is clearly required. This would delve into the different approaches currently existing which tackle this same problem. 

- There clearly exists a research gap which this paper correctly tries to tackle, but this is not properly explained. The above mentioned State-Of-The-Art section is important as it would then allow the authors to clearly show how they addressing the current research gap. 

- Figure 7 - It is not clear within the description in the text or within this figure how the Model is translated from a 3 Dimensional form to a 2 Dimensional form. This description needs to be greatly improved to detail (with various examples) the method by which this can be achieved. 

- It is not at all clear how Algorithm 2 would result in the tool path which is shown in the figure (unlabeled) at line 252. This needs to be explained further and in detail. 

- It is unclear if an actual evaluation of the work has been carried out. A section should be included which details exactly the result of an algorithm and compares it to a traditional approach. This could show the improvements (in terms of time for example, or computation) and also very importantly the error in the proposed algorithm in achieving the desired shape.

- Further to the above comments the paper needs to be clearly proof read as there are comments such as line 124: (cite xx – from my thesis)

- Not all figures are referenced in the text, and furthermore it is typical that figures are referred to in the text before the figure is included. 

Author Response

Dear Reviewer,

I thank you for your valuable comment. We have positively considered your comments and extensively revised the article. 

For detail point out;

- A section which clearly explains the State-Of-The-Art in this field is clearly required. This would delve into the different approaches currently existing which tackle this same problem. 

- There clearly exists a research gap which this paper correctly tries to tackle, but this is not properly explained. The above mentioned State-Of-The-Art section is important as it would then allow the authors to clearly show how they addressing the current research gap. 

=> Initially State-Of-The-Art was not part of the template for MDPI. However, we found your comment valuable and we included this section in the revised version. 

- Figure 7 - It is not clear within the description in the text or within this figure how the Model is translated from a 3 Dimensional form to a 2 Dimensional form. This description needs to be greatly improved to detail (with various examples) the method by which this can be achieved. 

= > Revised and extensively improved

- It is not at all clear how Algorithm 2 would result in the tool path which is shown in the figure (unlabeled) at line 252. This needs to be explained further and in detail. 

=> It is true; the image doesn’t clearly represent the algorithm. Detailed information and modification is applied and improved.

- It is unclear if an actual evaluation of the work has been carried out. A section should be included which details exactly the result of an algorithm and compares it to a traditional approach. This could show the improvements (in terms of time for example, or computation) and also very importantly the error in the proposed algorithm in achieving the desired shape.

=> In this article, the aim is to develop a concept to generate tool path for numerically controlled machines using image processing techniques. The algorithm takes CAD model (which is considered as ground truth) and generates tool path in Cartesian coordinate system. However, the approach we applied is one of the main contribution of this article. In addition, we translate the Cartesian coordinate points into machine readable language like G-code. Finally we visualized the generated path and discussed with respect to the original model. Actual realization using real machine can be considered in the future work. Our result is accurate and reliable regarding cost and time.

- Further to the above comments the paper needs to be clearly proof read as there are comments such as line 124: (cite xx – from my thesis)

=> This was a forgotten to do list before the submission and  now corrected.

- Not all figures are referenced in the text, and furthermore it is typical that figures are referred to in the text before the figure is included. 

=> This issue is now totally resolved.

In general, we have checked grammar and style problems and extensively revised all sections. 

Kind regards, and thank you for your consideration.

Authors

Reviewer 3 Report

Interesting piece of research with average industrial impact.

The references are well selected but more could have been used.

I propose its acceptance as is. 

Author Response

Dear reviewer, 

We would like to thank you for your valuable feedback. We have considered your suggestion regarding grammar and methodologies, we have improved the article. 

Kind regards and thank you for your consideration,

Authors

Reviewer 4 Report

The article presents the original approach of the author to the automatic generation of the numerical processing code, however, both the title and the list of articles raises objections

Comment 1

Artcle title

 Automated tool path generation using image 4 processing techniques It suggests a much wider scope of the solution. This will concern the automatic generation of G-code based on the image analysis of the object.

For example “In article Indication the Machining Area with the Robot's Camera Using  Book Series: Applied Mechanics and Materials”   Volume: 186   Pages: 50-57   Published: 2012” G-code processing control was developed based on image analysis, similarly at work “Machining with Image Recognition Using Industrial Robot", Applied Mechanics and Materials, Vol. 186, pp. 50-57, 2012”

Comment 2

The issue of using stl files to generate machining control code was analyzed in the “Raster milling tool-path generation from STL files. Rapid Prototyping Journal; 2006, Vol. 12 Issue 1, p4-11 where generation of machining control files using ball milling cutters was achieved.

Comment 3

I suggest extending the list of references with the presented items simultaneously with changing the title of the topic to clarify these doubts

The article contains a number of editing errors which should be corrected

Comment 4

Figure 1 not described before it presentation In text

Figure 2 not described in text

Table 1 is not  described in text and presented data not analysed in article

Figure 4 not described in text in this figure not described a) and c) parts

Comment 5

An Algorithm 1 is presented in the text I propose show it as table 2 or picture Please describe in which language it was developed (I suppose that MatLab) but it is not clearly presented

That same An Algorithm 2 is presented in the text tekstu  I propose show it as picture because was presented too the graphics

Comment 6

There is no analysis of the accuracy of the proposed method at work

In relation to the identified defects both substantive and editorial I propose a major review of article

Author Response

Comment 1

Artcle title

 Automated tool path generation using image 4 processing techniques It suggests a much wider scope of the solution. This will concern the automatic generation of G-code based on the image analysis of the object.

For example “In article Indication the Machining Area with the Robot's Camera Using  Book Series: Applied Mechanics and Materials”   Volume: 186   Pages: 50-57   Published: 2012” G-code processing control was developed based on image analysis, similarly at work “Machining with Image Recognition Using Industrial Robot", Applied Mechanics and Materials, Vol. 186, pp. 50-57, 2012”

=> In this work, it is aimed to present a reliable and automated method to generate tool path for numerically controlled machining tools slicing 3D models into 2D layers. Each layer is discretized and converted in a binarized image. After that a start point is selected, a tool path is generated in Cartesian coordinates between the image contiguous pixels and finally translated into machine language like G-code. 

=> Therefore, it is not using image for processing but using image processing techniques such as slicing, binarizing, subtraction, filtering, smoothening and pixel to 'mm' transformation. In the revised article the story and approaches are well described. We believe it will improve the ambiguity. The references that you specified are relevant and we have refered it.  

Comment 2

The issue of using stl files to generate machining control code was analyzed in the “Raster milling tool-path generation from STL files. Rapid Prototyping Journal; 2006, Vol. 12 Issue 1, p4-11 where generation of machining control files using ball milling cutters was achieved.

=> That is true, however, our primary intention is not to analyze but to use STL due to its advantage such as memory size, data arrangement (e.g. vertices, normals and facets). We consider the paper for state-of-the-art. 

Comment 3

I suggest extending the list of references with the presented items simultaneously with changing the title of the topic to clarify these doubts

The article contains a number of editing errors which should be corrected

=> Appreciated and considered.

Comment 4

Figure 1 not described before it presentation In text

Figure 2 not described in text

Table 1 is not  described in text and presented data not analysed in article

Figure 4 not described in text in this figure not described a) and c) parts

=> Appreciated and considered.

Comment 5

An Algorithm 1 is presented in the text I propose show it as table 2 or picture Please describe in which language it was developed (I suppose that MatLab) but it is not clearly presented

That same An Algorithm 2 is presented in the text tekstu  I propose show it as picture because was presented too the graphics

=> Appreciated and considered.

Comment 6

There is no analysis of the accuracy of the proposed method at work

=> This article does not provide quantitative result regarding accuracy. However, it is described using graphs and plots. We discuss the accuracy with respect to surface quality which is compared with the original model (3D Model). We believe the revised version presents and discusses quality and reliability of our approach. 

In relation to the identified defects both substantive and editorial I propose a major review of article

=> Following your suggestion, we did an extensive revision regarding technical and grammatical issues.

Thank you very much for your consideration

Authors

Round 2

Reviewer 2 Report

The comments which had initially been placed have been adequately addressed. 

Reviewer 4 Report

My reservations concern only incomplete documentation of the list of references.

Please provide all data for the following publications
[5] S. M. LaValle, “Rapidly-exploring random trees: A new tool for path planning,” ???????, 1998.
[21] X. Qu and B. Stucker, “Raster milling tool-path generation from STL files,” Rapid Prototyping Journal, ??????? Jan. 2006.
[22] M. Hassaballah,Aly A. Abdelmgeid, Hammam A. Alshazly, “Image Features Detection, Description and Matching.” ??????
[23] M. Eragubi, “Slicing 3D CAD Model in STL Format and Laser Path Generation,” IJIMT, ?????, 2013.
[24] M. Vatani, A. R. Rahimi, and F. Brazandeh, “An Enhanced Slicing Algorithm Using Nearest Distance Analysis for Layer Manufacturing,” ?????? vol. 3, no. 1, p. 6, 2009.
[25] M. Szucki and J. Suchy, “A voxelization based mesh generation algorithm for numerical models used in foundry engineering MichaÅ‚ Szucki, Józef S. Suchy.,” MAFE, vol. 38, no. 1, p. 43, 2012.

After imporóve this mistakes article may be published

Author Response

Dear esteemed reviewer, 

We would like to appreciate for your valuable comment again. We have resolved the issue related to bibliography. However, we removed one reference since we could not find detailed information. 

Therefore, we would like to request your approval for publication. 

Kind regards, 

Authors,